# Elderly Patients and Insect Venom Allergy: Are the Clinical Pictures and Immunological Parameters of Venom Allergy Age-Dependent?

**DOI:** 10.3390/vaccines12040394

**Published:** 2024-04-09

**Authors:** Robert Pawłowicz, Andrzej Bożek, Anna Dor-Wojnarowska, Marta Rosiek-Biegus, Agnieszka Kopeć, Małgorzata Gillert-Smutnicka, Małgorzata Sobieszczańska, Marita Nittner-Marszalska

**Affiliations:** 1Clinical Department of Internal Medicine, Pneumology and Allergology, Wroclaw Medical University, Chałbińskiego 1a, 50-368 Wrocław, Poland; anna.dor-wojnarowska@umw.edu.pl (A.D.-W.); marta.rosiek-biegus@umw.edu.pl (M.R.-B.); agnieszka.kopec@umw.edu.pl (A.K.); marita.nittner-marszalska@umw.edu.pl (M.N.-M.); 2Clinical Department of Internal Diseases, Dermatology and Allergology, Medical University of Silesia, Sklodowskiej 10, 41-800 Zabrze, Poland; andrzej.bozek@sum.edu.pl; 3IVth Pediatric Departament, Provincial Specialist Hospital, ul. Koszarowa 5, 51-149 Wrocław, Poland; 4Clinical Department of Geriatrics, Wroclaw Medical University, Chałbińskiego 1a, 50-368 Wrocław, Poland; malgorzata.sobieszczanska@umw.edu.pl

**Keywords:** Hymenoptera venom allergy (HVA), immunotherapy, elderly people, immunological parameters

## Abstract

Insect venom is one of the most common triggers of anaphylaxis in the elderly population. Venom immunotherapy (VIT) remains the only treatment for Hymenoptera venom allergy (HVA). However, little is known about the differences in indication for VIT in the group of patients aged 60 years and older. The objective of this study was to assess the clinical and diagnostic differences of HVA in elderly patients. The study compared data from patients aged ≥ 60 (N = 132) to data from patients aged from 11 to 60 years (N = 750) in terms of HVA severity, comorbidities, and immunological parameters, namely, intradermal testing (IDT), specific IgE (sIgE) levels against extracts and major allergenic molecules, and serum tryptase level (sBT). The severity of systemic HVA (I–IV Müller scale) did not differ between adults and seniors. However, the severity of cardiovascular reactions (IV) increased with age, while the frequency of respiratory reactions (III) decreased. No differences were found in the immunological parameters of sensitization IDT, venom-specific IgE concentrations, or sIgE against Api m 1, 2, 4, 5, and 10 between patients below and above 60 or 65 years of age. Differences were noted for sIgE against Ves v1 and Ves v5; they were higher and lower, respectively, in seniors. In the seniors group, sBT levels were higher. Elevated tryptase levels, along with the aging process, can represent a risk factor within this age category. Nevertheless, advanced age does not influence the immunological parameters of immediate HVA reactions, nor does it impact the diagnosis of HVA.

## 1. Introduction

While allergies are present throughout the entire human lifespan, their prevalence, severity, and triggers vary across different age groups. Given the increasing proportion of seniors in the general population and the rising trend of allergies in recent decades, addressing the diagnosis and treatment of allergic diseases in the older population has become increasingly important [1,2,3].

Hymenoptera venom allergy (HVA) holds a unique position among allergies. It stands as a prominent cause of anaphylaxis [4,5]. The estimated prevalence of HVA in European population studies ranges from 7.5% to 8.9% [5,6,7]. Symptoms of HVA can manifest at any age, spanning a range of severity from mild local reactions to life-threatening systemic responses (HVA-SYS) [7].

Age-related comorbidities are potential contributors to the heightened severity of HVA anaphylaxis [3,8,9,10]. Concurrent cardiovascular diseases could impair cardiovascular compensatory mechanisms during anaphylaxis [11]. Furthermore, consideration must be given to medication intake, particularly beta-blockers and ACE inhibitors, which may hinder an individual’s capacity to compensate and recover during anaphylaxis [12]. Another factor affecting the course of HVA in the elderly is the altered reactivity of mast cells due to aging-related factors [13].

Current treatment guidelines for HVA do not distinguish between different age groups [14]. Epinephrine is recommended for the management of anaphylactic reactions regardless of age. Similarly, venom immunotherapy (VIT) is universally recommended for the prophylactic treatment of HVA. Qualification criteria for VIT include clinical factors, positive venom allergen skin test outcomes, and/or positive specific IgE antibodies to whole-venom extract. Nevertheless, it remains unexplored as to whether older age influences these parameters in HVA patients and whether age-related changes in the immunological response, termed immunosenescence, impact the diagnosis of HVA and the qualification process for VIT [15].

The objective of our study is to investigate whether advanced age influences the severity of HVA and the outcomes of diagnostic tests assessing the immunological response to insect venom. Through our collected data, we aim to confirm potential clinical and diagnostic distinctions that might arise in elderly HVA patients compared to their younger counterparts.

## 2. Methods

### 2.1. Database and Cohort

The study was conducted on 882 adult patients who presented symptoms of systemic allergic reactions following insect stings (HVA-SYS). The study group consisted of patients undergoing diagnostics and treatment at the Clinic of Internal Medicine and Allergology, a reference center for the care of patients with allergies to Hymenoptera venom. This group mainly included patients from southwestern Poland. Exclusion criteria were pregnancy, lactation, severe mental disorders, and lack of patient consent. Patients were categorized into two groups: those younger than 60 years (N = 750) and those aged 60 years and older (N = 132). The assessment of patients over 60 years of age was intended to observe the phenomenon of insect venom allergy in seniors, looking for differences from younger patients. The National Health Fund in Poland considers people over 60 as seniors. However, in some European countries, the age limit for seniors is 65, which resulted in the additional classification of the 65+ group and their assessment in the presented study [16].

We analyzed the following aspects in all groups:•The severity of reactions was determined using the Müller scale (grades I–IV). The assessment of the severity of HVA reactions is based on the four-grade Müller scale: grade I (urticaria, itching, malaise); grade II (grade I symptoms + angioedema, abdominal pain, nausea, tightness in the chest); grade III (grade I and II symptoms + dyspnea, respiratory symptoms); and grade IV (grade I, II, and III symptoms + cardiovascular symptoms, loss of consciousness, cyanosis).•The species of the causative insect.•The presence of accompanying diseases and their treatment.

Confirmation of venom allergy was established based on positive results of the conducted tests: skin tests with venom and/or serological tests (positive sIgE result against venom extract). Serum tryptase levels were measured in all patients. The study was approved by the Ethics Committee of the Medical University of Wroclaw.

### 2.2. Diagnostic Procedures

Subcutaneous tests were performed on the volar surface of the forearm with a volume of 0.02 mL of an aqueous solution of wasp and bee venom extract (Pharmalgen ALK-Abello, Hørsholm, Denmark). The test began with venom dilution of 0.01 µg/mL, and the venom concentration was increased tenfold every 15–20 min until a positive test result was obtained or the threshold venom concentration of 1 µg/mL was reached. The testing technique followed the recommendations of EAACI [17]. A positive result was defined as an increase in wheal diameter of at least 3 mm accompanied by redness. Negative control tests were conducted using 0.9% NaCl, and positive control tests were performed using histamine.

Serum venom-specific IgE tests (sIgE): to evaluate the sIgE with the venom extract, determination of the sIgE BV extract was performed using the ELISA method. The cut-off point for the positive result was 0.35 kUA/L.

Component diagnosis: determination of the sIgE BV allergen components was performed using the ImmunoCAP system measurements (Thermo Fisher Scientific Inc., Göteborg, Sweden) using the Phadia100 device (Thermo Fisher Scientific Inc., Göteborg, Sweden) via the ELISA method. In addition to the standard curve (ImmunoCAP Specific IgE Calibrators No: 10-9460-01), controls were performed during each analysis (ImmunoCAP Specific IgE No: 10-9462-01). The following components in bee and wasp venom have been determined: Api m 1 Phospholipase A2, Honey bee, No: 10 14-4987-01; Api m 2 Hyaluronidase, Honey bee, No: 14-6014-01; Api m 3 Acid phosphatase, Honey bee, No: 14-6015-01; Api m 5 Dipeptidyl peptidase, Honey bee, No: 14-6016-01; Api m 10 Icarapin, Honey bee, No: 14-6004-01; Ves v 1 Phospholipase A1, Common wasp; No: 14-4995-01; Ves v 5 Common wasp; No: 14-4992-01

Baseline serum mast cell tryptase concentration (BTC) was measured in blood samples collected at least one month after the most recent systemic allergic reaction. The processing was carried out following the guidelines provided by the manufacturer of the tryptase assay. Serum tryptase concentrations were determined using the ImmunoCAP Tryptase kit. According to the manufacturer’s specifications, the inter-assay variability for tryptase levels ranging from 1 to 10 mg/L is below 5%. The upper 95th percentile for healthy non-allergic individuals is 11.4 mg/L.

### 2.3. Statistical Analysis

Data analysis was undertaken using Statistica 13 software (TIBCO Software Inc. version 13.3.721.0). A Shapiro–Wilk test was performed to check the normal distribution of the sample. Due to the fact that the studied parameters were not normally distributed, non-parametric tests were used. The data were expressed as median (Me) and the lowest (Min) and highest (Max) parameter values. A Mann–Whitney test or Ch^2^ test were performed in order to compare two independent samples. For dependent samples, Wilcoxon or McNemar tests were performed. Correlation between variables was analyzed by means of a Spearman test. A *p* < 0.05 was regarded as a statistically significant value for all tests.

## 3. Results

Data obtained from 882 subjects were analyzed. The demographic characteristics of the study cohort are presented in Table 1. The older seniors group constituted 15% of the entire cohort of seniors (N = 132). Among them, there were 69 younger seniors with a median age of 62 (60–64), accounting for 7.8% of the entire cohort, and 63 older seniors with a median age of 68 (65–79), accounting for 7.2% of the entire cohort. The age distribution among younger patients and the elderly is depicted in Figure 1. The number of women was higher in the elderly group than in the younger group (*p* = 0.03). Across all age groups, symptoms of HVA were predominantly triggered by wasp stings. Cardiovascular diseases, diabetes, and thyroid disease were more prevalent in the elderly group (*p* < 0.001). No significant differences between the younger and elderly groups were observed with respect to atopic diseases (Table 1).

### 3.1. Severity of HVA Reactions in Young and Elderly Patients

Non-significant differences in the severity of HVA reactions between patients under and over 60 were shown. The overall incidence of I+II, III, and IV HVA reactions (Mueller’s scale) did not differ in the patients under and over 60 (*p* > 0.05).

Similarly, there were no differences in the frequency of non-life-threatening reactions (SYS I+SYS II) vs. life-threatening reactions (SYS III + SYS IV) between the younger and older groups.

Non-life threatening reactions amounted to 17.9% in younger patients (114 vs. 636) and 12.8% in the elderly group (15 vs. 117) (*p* > 0.05).

Viewed against the combined three grades of HVA reactions (SYS I+II+III), reactions did not occur with greater frequency in the younger (N = 335/750; 44.6%) group than in the elderly group (N = 65/132; 49%) (*p* < 0.05) (Figure 2).

It was found that the percentage of patients with cardiovascular symptoms of HVA (SYS IV) increased with age; the percentage of such patients assessed at ten-year age intervals increased from age 40 onwards (*p* < 0.001) (Figure 2). Conversely, the frequency of III reactions decreased with age in consecutive ten-year age intervals (*p* = 0.005).

### 3.2. ACE and Beta Inhibitors and the Severity of HVA

The intake of beta-blockers and ACE inhibitors and ARBs was higher in the elderly group than in the younger patients (for all *p* < 0.001). However, the use of beta-blockers and ACE inhibitors was not linked to a more severe course of allergic reactions following a sting in the elderly for both groups (*p* > 0.05).

### 3.3. sBT and Immunological Features Characterizing Elderly HVA Patients (IDT, Venom Specific sIgE)

In the senior age groups (over 60 and over 65 years of age), sBT levels were observed to be significantly higher than in the younger age group (*p* = 0.000) (Table 2; Figure 3). None of the subjects under examination had a diagnosis of mastocytosis at the time of qualification for the study. Patients with elevated levels of tryptase were selected for further investigations to ascertain the diagnosis of primary mast cell activation syndromes.

There were no significant differences in the results of IDT tests using insect venom between the elderly patient groups (over 60 and over 65 years of age) and the younger age group. Similarly, the results of cIgE and sIgE tests did not exhibit significant differences (Table 3) among the different age groups.

### 3.4. Molecular Diagnostic HVA (sIgE against Api m 1, 2, 4, 5, 10, Ves v 1, Ves v 5)

No differences were observed in the concentration of sIgE targeted against the main bee venom molecules between the two examined age groups. The assessment of sIgE against the Ves v 1 molecule revealed higher antibody concentrations in individuals >60 years of age (*p* = 0.004). Conversely, sIgE against Ves v 5 were lower in individuals >60 years of age (*p* = 0.009) (Table 2).

## 4. Discussion

The purpose of our study was to assess the profile of HVA clinical symptoms and immunological reactivity to insect venom in a cohort of patients over 60 years of age. To our knowledge, ours is the first attempt at analyzing immunological HVA parameters in older age groups and one of the few analyses examining the distinct clinical features of HVA in the elderly population.

The species of insects responsible for stings in our geographical region are analogous to those in southern regions of Germany and Austria. Unfortunately, we do not have entomological data illustrating the species differences of stinging insects in southern Poland. Among those stung, wasps are the most common, while stings from hornets are rarest, which is consistent with data from the European registry published by M. Worm et al. [5].

The most common elicitors of insect allergy are bees, bumblebees, wasps, and hornets. The Vespula genus, particularly Vespula wasps, is responsible for a significant portion of allergic reactions in Europe. These wasps are carnivores and often come into contact with human populations, especially during outdoor activities like eating outdoors. Unlike the Dolichovespula genus, Vespula wasps are a common cause of allergic reactions due to their interactions with humans. The search results indicate that Vespa velutina, a wasp species originating from Asia, has been identified in Europe and is associated with anaphylactic reactions. This species was first discovered in Europe in southern France around the end of 2005 and has since spread to various regions. Studies have shown that individuals who have been stung by Vespa velutina may experience systemic reactions, with some patients exhibiting specific IgE responses to components of the venom. The presence of cross-reactive antigens in Vespa velutina suggests potential challenges in immunotherapy due to the lack of specific venom available for treatment [18].

On the basis of our result, the occurrence of cardiovascular symptoms of HVA tends to increase linearly with age, while the frequency of respiratory symptoms (typical for III HVA) decreases with advancing age. The study did not show any differences in the immunological response to venom allergens in people above and below 60 years of age. Therefore, age does not affect the qualification for venom immunotherapy.

From a clinical perspective, two sets of findings seem to be significant. The first one includes outcomes that mark a distinction between the older and the younger HVA-SYS patients. Thus, we found that with aging, HVA reactions tend to involve cardiovascular symptoms more often. This tendency shows from age 40 onward. HVA-SYS IV reactions in the age groups over 60 and over 65 were not found to be more frequent compared with the younger study group, which might be accounted for by the linear character of this relationship and/or by too small a number of older patients participating in the study. Nevertheless, due to the increased danger of cardiovascular complications, older age is a risk factor that has to be reckoned with in the course of HVA. The age–severity relationship in HVA has been studied before [1,2,3,19]. Rueff et al. suggested that older age in HVA patients is a factor associated with cardiac arrest, anaphylactic shock, and loss of consciousness [3]. In a study conducted on 324 people subjected to a sting challenge, Van der Linden et al. demonstrated a correlation between age and the severity of reactions after a sting challenge [19]. Arzt et al. reported that individuals aged over 40 are at a greater risk of severe systemic insect sting reactions [20]. The European Anaphylaxis Registry points out not only a correspondence of the severity of anaphylactic reactions with age but with more frequent concomitance of cardiovascular symptoms as well [2]. Still, unlike the others, in a study on food allergy, Pumphrey et al. did not find a link between age and intensification of anaphylactic reactions [21]. Thus, in view of the above, a question may be asked as to whether the presented age-related correspondences are limited to HVA anaphylaxis. Finding an answer to the question may pose problems as a comparative analysis of the results of studies investigating the impact of aging on the clinical picture of anaphylaxis is rendered difficult due to the distinct etiologies of anaphylaxis and the non-uniform methodologies adopted by the researchers, for instance, diverse definitions of “severe anaphylaxis” and different classifications adopted to assess the reactions.

In whole insect allergic groups, there are more patients with elevated baseline serum tryptase (BTC) levels due to clonal mast cell disorders, especially in cases of severe Hymenoptera venom allergy. These disorders, characterized by abnormal mast cell proliferation and activation, increase the risk of severe anaphylactic reactions. The presence of clonal mast cell diseases like systemic mastocytosis and monoclonal mast cell activation syndrome is particularly significant in patients who suffer from venom-induced anaphylaxis.

Considering the severity issue of HVA reactions, we indicate two factors that may put older age individuals at a disadvantage. The former relates to elevated sBT. In our study, BTC levels were higher in the elderly groups compared to the younger group. Higher BCT levels found in the older-age groups correspond to reports on the aging-related deterioration of mast cells’ functions; the mast cells’ functional impairment may be a potential risk factor of more severe HVA reactions in older age [13]. A few studies evaluating risk factors for anaphylaxis have stressed an association between sBT levels, clonal mastocyte disorders, and higher frequency and severity of sting-induced anaphylaxis [22,23,24,25]. The latter refers to aging-related comorbidities, especially cardiovascular diseases, which may have a negative impact on the prognosis of the severity of HVA reactions in old age. The severity of HVA symptoms related to cardiovascular conditions may be additionally affected by the ACE inhibitors and beta-blockers used in routine treatment of the illnesses, which has been reported by both experimental and observation studies [1,2]. While cardiovascular diseases involving taking ACE inhibitors and beta-blockers were more frequent in our elderly study group, we did not find correspondence between the use of these medications and aggravation of HVA symptoms. Likewise, some other researchers have failed to demonstrate a negative influence of ACE inhibitors and beta-blockers on the severity of HVA reactions [20,26].

A comparative analysis showed no significant differences regarding the parameters of immunological reactivity to insect venom; neither the magnitude of IDT nor the serological indexes of sensitization were found to change at older age. Hence, the process of diagnosing HVA and, if recommended, qualifying patients for venom immunotherapy, does not seem to be affected by aging. Data on sensitization in the elderly are contradictory. Some clinical studies, supported by epidemiological data, have demonstrated an age-related decrease in serum total IgE and sIgE levels against inhalative allergens in older patients. Assessments of serum total IgE conducted in younger and older subjects without any allergic diseases have shown considerably lower levels in the elderly [27,28]. Other studies, in turn, indicate that immunosenescence does not affect the levels of sIgE in atopic patients [29,30].

The prevalence of HVA allergy in the elderly is unknown. In our study, elderly individuals constituted 15% of the HVA-SYS cohort. Considering the fact that our group was predominantly made up of III and IV HVA-SYS subjects, the percentage of patients with milder manifestations of this allergy may be higher.

The limitations of this study include its single-center nature and the small number of patients in the group over 65 years old, which hinders the assessment in subsequent 10-year age intervals. A further limitation could be the low representation of patients under 18 years of age. The age disproportion, however, reflects the unequal frequency of insect venom allergies, especially severe cases, in children and adolescents (rarer) compared to adults (more common).

## 5. Conclusions

To conclude, the potential for increased severity of HVA reactions with age should be taken into consideration. Elevated tryptase levels, along with the aging process, might pose a risk factor within this age category. Nevertheless, advanced age does not influence the immunological parameters of immediate HVA reactions, nor does it impact the diagnosis of HVA.

## Figures and Tables

**Figure 1 vaccines-12-00394-f001:**
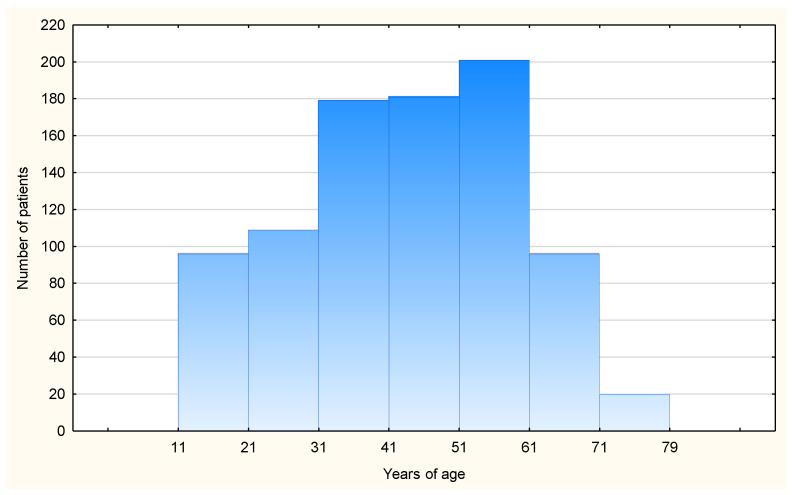
The study patients’ age distribution.

**Figure 2 vaccines-12-00394-f002:**
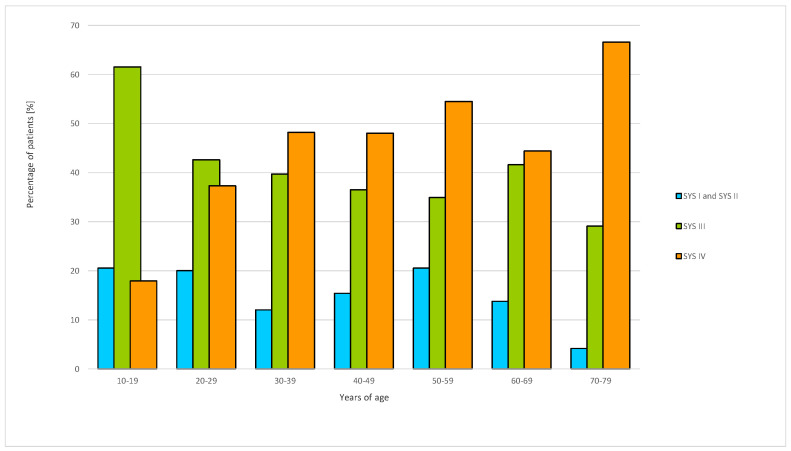
Severity of HVA reactions according to Mueller’s scale at ten-year age intervals. SYS—systemic reaction grade I, II, III, and IV.

**Figure 3 vaccines-12-00394-f003:**
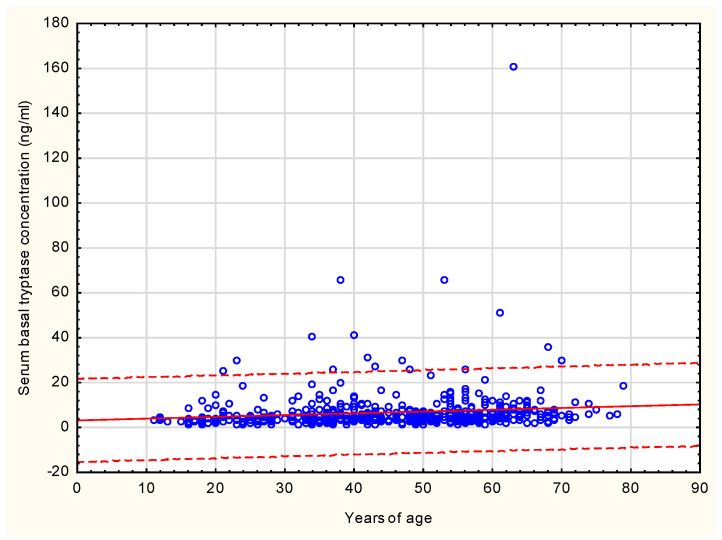
Correlation between serum basal tryptase concentration and patients’ age.

**Table 1 vaccines-12-00394-t001:** The demographic and clinical characteristics of the study groups.

Parameters	Patients <60 Years of Age N = 750	Patients ≥ 60 Years of Age N = 132	*p*
Gender women/men	395/355	88/44	*p* = 0.03
Age [years] median (min–max)	40 (11.0–59.0)	64 (60.0–79.0)	
Ischemic heart disease n (%)	12 (1.60)	13 (9.84)	*p* = 0.00
Arterial hypertension n (%)	150 (20.00)	76 (57.57)	*p* = 0.00
Diabetes n (%)	7 (0.93)	18 (13.63)	*p* = 0.00
Allergy n (%)	124 (16.53)	27 (20.45)	*p* = 0.27
Thyroid diseases	68 (9.06)	27 (20.45)	*p* = 0.00
Beta-blockers	41 (5.46)	32 (24.24)	*p* = 0.00
ACEi	52 (6.93)	28 (21.21)	*p* = 0.00
ARBs	39 (5.2)	25 (18.9)	*p* = 0.00

N—number of patients; %—percentage of patients; ACEi—angiotensin converting enzyme inhibitors; ARBs—angiotensin II receptor blockers; p—chi^2^ Pearson’s test.

**Table 2 vaccines-12-00394-t002:** Immunological parameters of HVA in the study groups.

Parameters	Patients <60 Years of Age N = 750	Patients ≥ 60 Years of Age N = 132	*p*
Bee venom allergy	211 (28.14%)	35 (26.52%)	*p* = 0.78
Wasp venom allergy	514 (68.53%)	94 (71.21%)	
Bee and wasp venom allergy	25 (3.33%)	3 (2.27%)	
Total IgE median (min-max)	63.50 (5.44–1670.00)	59.10 (4.20–4590.00)	*p* = 0.49
Insect specific IgE mean (min–max) sIgE Api m 1	5.01 (0.00–64.90)	6.39 (0.00–67.80)	*p** = 0.66
Insect specific IgE mean (min-max) sIgE Api m 2	1.59 (0.00–23.20)	1.98 (0.00–20.87)	*p** = 0.33
Insect specific IgE mean (min-max) sIgE Api m 5	0.48 (0.00–3.00)	0.71 (0.00–7.77)	*p** = 0.07
Insect specific IgE mean (min-max) sIgE Api m 10	2.98 (0.00–47.70)	4.98 (0.01–86.30)	*p** = 0.15
Insect specific IgE ean (min-max) sIgE Ves v1	5.89 (0.00–100.00)	9.50 (0.00–100.00)	*p** = 0.004
Insect specific IgE mean (min-max) sIgE Ves v5	7.23 (0.00–100.00)	6.91 (0.00–100.00)	*p** = 0.009
IDT (diameter) [mm] median (min-max)	11.00 (0.00–24.00)	11.00 (0.00–20.00)	*p* = 0.65
IDT (area) [mm^2^] median (min-max)	94.00 (0.00–1838.00)	94.00 (0.00–307.00)	*p* = 0.67
sBT [ng/mL] median (min-max)	4.31 (1.00–65.80)	6.01 (1.98–16,100)	*p* = 0.000

N—number of patients; %—percentage of patients; *p*—Pearson’s chi^2^ test; *p**—Mann–Whitney U test; IDT—intradermal test; sBT—serum basal tryptase concentration.

**Table 3 vaccines-12-00394-t003:** The demographic and clinical characteristics of two subgroups of elderly patients (11–64 years old and 65 years old and older).

Parameters	The Elderly Patients ≤64 Years of Age N = 819	The Elderly Patients ≥ 65 Years of Age N = 63	*p*
Age [years] median; (min-max)	42 (11–64)	68 (65–79)	
Gender women/men	380/46	19/30	*p* = 0.001
Insect specific IgE median; (min-max)	4.1 (0.00–101.0)	3.0 (0.0–91.8)	*p* > 0.05
Total IgE median; (min-max)	65.7 (4.2–4590.0)	55.5 (9.1–967.0)	*p* > 0.05
IDT (diameter) [mm] median; (min-max)	11.0 (0.0–24.0)	12.0 (0.0–20.0)	*p* > 0.05
IDT (area) [mm^2^] median; (min-max)	94.0 (0.0–1838.0)	112.0 (0.0–240.0)	*p* > 0.05

N—number of patients; *p*—chi^2^ Pearson’s test; IDT—intradermal test.

## Data Availability

Data available on resonable request due to restrictions (e.g., privacy, legal or ethical reasons).

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
