# Peer review of "Elderly Patients and Insect Venom Allergy: Are the Clinical Pictures and Immunological Parameters of Venom Allergy Age-Dependent?"

_vaccines, 2024, doi:10.3390/vaccines12040394_

Round 1

Reviewer 1 Report

Comments and Suggestions for Authors

The authors conclude that serious reactions with a cardiovascular component depend on age, as well as tryptase levels.

Quintela et al in this article relate age to tryptase: Serum total tryptase concentrations are particularly dependent on age (Gonzalez-Quintela A, Vizcaino L, Gude F, Rey J, Meijide L, Fernandez-Merino C, Linneberg A, Vidal C. Factors influencing serum total tryptase concentrations in a general adult population. Clin Chem Lab Med. 2010 May;48(5):701-6.)

Additional Comments:

The cut-off or selection age of 60 years, what is it based on? Are a 59-year-old patient or a 61-year-old patient classified into different groups? What is the justification for classifying them into different groups, do they really belong to different groups?

By the way, the possibility that HVA reactions increase with age should be taken into account, which would reinforce the role of immunotherapy with venom  in this age group.

Author Response

Thank you for the thorough analysis of our article and the valuable comments that have been raised. We will endeavor to incorporate all suggestions and corrections to enhance the quality of the publication. Should there be any specific questions or clarifications needed, please feel free to reach out, as we are more than happy to address any additional inquiries. Once again, we appreciate your time and valuable feedback.

  1. The authors conclude that serious reactions with a cardiovascular component depend on age, as well as tryptase levels.

Quintela et al in this article relate age to tryptase: Serum total tryptase concentrations are particularly dependent on age (Gonzalez-Quintela A, Vizcaino L, Gude F, Rey J, Meijide L, Fernandez-Merino C, Linneberg A, Vidal C. Factors influencing serum total tryptase concentrations in a general adult population. Clin Chem Lab Med. 2010 May;48(5):701-6.)

Ad1.  Thank you for pointing out this important article. We will add it to the literature. Additionally,  an independent factor influencing the severity of the reaction may be the age itself, as indicated by observations, including those by Arzt et al. (Hymenoptera stings in the head region induce impressive, but not severe sting reactions L. Arzt, D. Bokanovic, I. Schwarz, C. Schrautzer, C. Massone, M. Horn, W. Aberer, G. Sturm Allergy 2016 https://doi.org/10.1111/all.12967. Our results confirm this in the senior population. In practice, both factors should be taken into account when qualifying patients with HVA to assess the prognosis of subsequent reactions and consequently to qualify for VIT.

  1. The cut-off or selection age of 60 years, what is it based on? Are a 59-year-old patient or a 61-year-old patient classified into different groups? What is the justification for classifying them into different groups, do they really belong to different groups?

Ad2 Thank you for addressing the issue of the age threshold of 60+ adopted by us as a qualification for the group of individuals. The assessment of patients over 60 years of age aimed to observe the phenomenon of insect venom allergy in seniors, looking for differences from younger patients. The National Health Fund in Poland considers people over 60 as seniors. However, in some European countries, the age limit for seniors is 65, which resulted in the additional classification of the 65+ group and their assessment in the presented study (Rosenthal T., Naughton B., Williamsa M.: Geriatria, CZELEJ, Lublin, 2009).

Reviewer 2 Report

Comments and Suggestions for Authors

Dear Authors, The work is interesting, I have read it carefully. I have a series of comments and observations to convey to the authors, with the aim of improving the manuscript.

Title

It does not reflect the content of the work presented. Wrong in your writing. Missing question mark?

 In this sense, a linguistic review of the entire MS would be recommended to improve the presentation of the content and its reading.

Abstract: Acronyms used must also be defined in this section.

The abstract should be a single paragraph, not divided into subsections. Review the style of the Journal, especially in the references section. Which must be adequate and homogeneous. The following main subsections could be numbered in headings, to facilitate the reader's reading and understanding of the manuscript quickly.

And the keywords after the abstract?

Introduction

The first paragraph must be supported by some specific bibliographic reference.

It would be appropriate to cite the following work in the second paragraph:

Feás, X., Vidal, C., & Remesar, S. (2022). What We Know about Sting-Related Deaths? Human Fatalities Caused by Hornet, Wasp and Bee Stings in Europe (1994-2016). Biology, 11(2), 282. https://doi.org/10.3390/biology11020282

M & M

The patient selection criteria, which geographical area or hospital they come from, should be explained a little more.

The patients are divided into two groups, but later the group of those over 60 years of age is subdivided into two other groups as results and discussion are presented. There is no reference to this in M & M. It must be explained, as well as what the criteria for making this subdivision were. It's confusing.

Results

Results are presented in relation to the insect causing the allergy, but generally bee or wasp. It would be interesting to provide some more information about the hymenoptera insects in the area as well as if there is any in particular with a greater presence in the stings and developed allergies (bee, bumblebee, wasp, hornet... polistes, vespula, vespa...

Figure 3 and 4 must be redone. They have words in Polish. They are difficult to understand what is intended to be presented.

In tables, a single significant figure is sufficient.

Discussion

The presence of invasive insects is affecting allergies, see and cite the following works. The expansion of, for example, the Vespa vleutina in Poland is expected, and the authorities and especially allergists must take this into account. Comment on the above and cite the following works at this point.

Vidal, C. The Asian wasp Vespa velutina nigrithorax: Entomological and allergological characteristics. Clin. Exp. Allergy 202252, 489–498

At the end of the discussion, specifying the limitations of the present work would be appropriate.

A conclusion section would be appropriate.

Comments on the Quality of English Language

revising the English would improve the clarity of the manuscript

Author Response

Thank you for the thorough analysis of our article and the valuable comments that have been raised. We will endeavor to incorporate all suggestions and corrections to enhance the quality of the publication. Should there be any specific questions or clarifications needed, please feel free to reach out, as we are more than happy to address any additional inquiries. Once again, we appreciate your time and valuable feedback.

1.Title. It does not reflect the content of the work presented. Wrong in your writing. Missing question mark?

 In this sense, a linguistic review of the entire MS would be recommended to improve the presentation of the content and its reading.

Ad 1. Title. We apologize for the error in the manuscript editing that resulted in the lack of a question mark. The title has been corrected, and we hope it now reads correctly.

2. Abstract: Acronyms used must also be defined in this section.The abstract should be a single paragraph, not divided into subsections. Review the style of the Journal, especially in the references section. Which must be adequate and homogeneous. The following main subsections could be numbered in headings, to facilitate the reader's reading and understanding of the manuscript quickly. And the keywords after the abstract?

Ad 2. Abstract and keywords. The abbreviations used in the abstract have been defined. The abstract has been rewritten according to the journal's requirements, and keywords have been added.

3. Introduction. The first paragraph must be supported by some specific bibliographic reference. It would be appropriate to cite the following work in the second paragraph: 

Feás, X., Vidal, C., & Remesar, S. (2022). What We Know about Sting-Related Deaths? Human Fatalities Caused by Hornet, Wasp and Bee Stings in Europe (1994-2016). Biology, 11(2), 282. https://doi.org/10.3390/biology11020282

Ad 3. Introduction. In the first paragraph, a bibliographic reference has been added. Similarly, in the second paragraph, a suggested bibliographic reference has been added.

Aurich S, Dölle-Bierke S, Francuzik W, Bilo MB, Christoff G, Fernandez-Rivas M, Hawranek T, Pföhler C, Poziomkowska-GÈ©sicka I, Renaudin JM, Oppel E, Scherer K, Treudler R, Worm M. Anaphylaxis in elderly patients-data from the European anaphylaxis registry. Front Immunol. 2019;10:750.

Worm M, Francuzik W, Renaudin JM, Bilo MB, Cardona V, Scherer Hofmeier K, Köhli A, Bauer A, Christoff G, Cichocka-Jarosz E, Hawranek T, Hourihane JO, Lange L, Mahler V, Muraro A, Papadopoulos NG, Pföhler C, Poziomkowska-GÄ™sicka I, Ruëff F, Spindler T, Treudler R, Fernandez-Rivas M, Dölle S. Factors increasing the risk for a severe reaction in anaphylaxis: An analysis of data from The European Anaphylaxis Registry. Allergy. 2018;73:1322–1330.

Rueff et al. Predictors of severe systemic anaphylactic reactions in patients with Hymenoptera venom allergy: importance of baseline serum tryptase-a study of the European Academy of Allergology and Clinical Immunology Interest Group on Insect Venom Hypersensitivity. J Allergy Clin Immunol. 2009 Nov;124(5):1047-54. doi: 10.1016/j.jaci.2009.08.027

Feás X, Vidal C, Remesar S. (2022). What We Know about Sting-Related Deaths? Human Fatalities Caused by Hornet, Wasp and Bee Stings in Europe (1994-2016). Biology, 11(2), 282. https://doi.org/10.3390/biology11020282

4.The patient selection criteria, which geographical area or hospital they come from, should be explained a little more.

Ad 4. Methods and Materials: Thank you for your attention. The section "Database and cohort" has been corrected. The following content has been added: "The study group consisted of patients undergoing diagnostics and treatment at the Clinic of Internal Medicine and Allergology, a reference center for the care of patients with allergy to Hymenoptera venom. This group mainly included patients from southwestern Poland."

5. The patients are divided into two groups, but later the group of those over 60 years of age is subdivided into two other groups as results and discussion are presented. There is no reference to this in M & M. It must be explained, as well as what the criteria for making this subdivision were. It's confusing.

Ad 5. Thank you for addressing the issue of the age threshold of 60+ adopted by us as a qualification for the group of individuals. The assessment of patients over 60 years of age aimed to observe the phenomenon of insect venom allergy in seniors, looking for differences from younger patients. The National Health Fund in Poland considers people over 60 as seniors. However, in some European countries, the age limit for seniors is 65, which resulted in the additional classification of the 65+ group and their assessment in the presented study (Rosenthal T., Naughton B., Williamsa M.: Geriatria, CZELEJ, Lublin, 2009).

6. Results are presented in relation to the insect causing the allergy, but generally bee or wasp. It would be interesting to provide some more information about the hymenoptera insects in the area as well as if there is any in particular with a greater presence in the stings and developed allergies (bee, bumblebee, wasp, hornet... polistes, vespula, vespa...

Ad 6. Results. The species of insects responsible for stings in our geographical region are analogous to those in southern regions of Germany and Austria. Unfortunately, we do not have entomological data illustrating the species differences of stinging insects in southern Poland. Among those stung, wasps are the most common, while stings from hornets are rarest, which is consistent with data from the European registry published by Prof. M. Worm and her colleagues. 2014r We include this information in discussion section.

7. Figure 3 and 4 must be redone. They have words in Polish. They are difficult to understand what is intended to be presented.

Ad7. The figures have been removed as suggested by another reviewer.

8. Discussion. The presence of invasive insects is affecting allergies, see and cite the following works. The expansion of, for example, the Vespa vleutina in Poland is expected, and the authorities and especially allergists must take this into account. Comment on the above and cite the following works at this point.

Vidal, C. The Asian wasp Vespa velutina nigrithorax: Entomological and allergological characteristics. Clin. Exp. Allergy 202252, 489–498

 Ad 8. Discussion.  A paragraph regarding the possibility of stings by Vespa velutina has been added to the discussion.

The most common elicitors of insect allergy  are bees, bumblebees, wasps and hornets. The Vespula genus, particularly Vespula wasps, is responsible for a significant portion of allergic reactions in Europe. These wasps are carnivores and often come into contact with human populations, especially during outdoor activities like eating outdoors. Unlike the Dolichovespula genus, Vespula wasps are a common cause of allergic reactions due to their interactions with human. The search results indicate that Vespa velutina, a wasp species originating from Asia, has been identified in Europe and is associated with anaphylactic reactions. This species was first discovered in Europe in southern France around the end of 2005 and has since spread to various regions. Studies have shown that individuals who have been stung by Vespa velutina may experience systemic reactions, with some patients exhibiting specific IgE responses to components of the venom. The presence of cross-reactive antigens in Vespa velutina suggests potential challenges in immunotherapy due to the lack of specific venom available for treatment.

 An article by Vidal, C., The Asian wasp Vespa velutina nigrithorax: Entomological and allergological characteristics. Clin. Exp. Allergy 2022, 52, 489–498, has been quoted

9. At the end of the discussion, specifying the limitations of the present work would be appropriate. A conclusion section would be appropriate.

Ad9. A section on the limitations of the study and a summary have been added.

The limitations of the study include its single-center nature and the small number of patients in the group over 65 years old, which hinders the assessment in subsequent 10-year age intervals. Limiting work can also be considered as low representation of patients under 18 years of age. The disproportion of adults to youth and age, however, reflects the unequal frequency of insect venom allergies, especially severe cases in the group of children and adolescents (rarer) than in adults (more common).

To conclude the potential for increased severity of HVA reactions with age should be taken into consideration. Elevated tryptase levels, along with the aging process, might pose a risk factor within this age category. Nevertheless, advanced age does not influence the immunological parameters of immediate HVA reactions, nor does it impact the diagnosis of HVA.

Reviewer 3 Report

Comments and Suggestions for Authors

The authors submitted an interesting original research article, which deals with a relation between the immune system reaction to insect venom and age of patients. Since hypersensitive reactions against insect toxins can be very dangerous, the topic of this article is timely and important.

The research methods are adequate. The results are well presented in plots and are properly discussed. The authors cite relevant references.

Author Response

Thank you for the thorough analysis of our article and the valuable comments that have been raised. We appreciate your time and valuable feedback.

Reviewer 4 Report

Comments and Suggestions for Authors

The paper deals with a subject of undoubted interest (i.e.: clinical and immunological features of insect venom allergy in the elderly and validation of classical diagnostic tools in these patients) and deserves publication.

However, the manuscript appears to me rather immature and cannot be considered in the current form. Should the Authors be available to undertake a substantial revision, the manuscript may be re-assessed.

Here some suggestions.

1) Title. The English is overall acceptable (although, in a separate section of the reviewing form, I recommended a moderate editing). Therefore, I suppose that the grammar mistakes in it are due to simple lack of accuracy in the rush of the submission process. “Is the clinical pictures and immunological parameters ...age dependent”. Better: “Are the clinical picture and the immunological parameters...age dependent?

2) Abstract, Introduction and Methods – Database and cohort are acceptable. Here some example of minor inaccuracy (minor but still inaccuracy).

Lines 9-10: comparing patients...to data of patients. You cannot compare a patient to a datum, but you compare a patient to another patient or a set of data to another set of data.

Always leave a space after commas.

Line 20. ...might pose a risk factor... Something can pose a risk. Something else can represent a risk factor.

Lines 14 and 15. IVo...IIIo  The Latin numerals IV and III are usually used instead of 4° and 3°. No need for the small “o” beside them.

The above are only examples of the minor mistakes, mainly stylistic; scattered throughout the manuscript. From now on, I will not remark them one by one. But I warmly invite the Authors to go through the text with attention and emend them all.

3) 60/65 question. This is a major point. It is only source of confusion and adds nothing to the story you want to tell. A recommend removing these two sub-population (i.e: under-65 and over-65). Accordingly, emend all sections of the manuscript.

4) Line 76: Diagnostic procedure. The skin test methodology is described poorly. Moreover, it seems to me that there is a certain confusion with the concentration of the allergen. At line 78, it reads 1 mg/ml (wrong). At line 80, it reads 10-4 g/l. If I am not mistaking, it means 100 micrograms/l (wrong). The usual concentration is 100 micrograms/ml. I recommend improving much this sub-section. The Authors may found inspiration in Corallino M. et al. J Clin Nurs. 2007:16;1256-64. Or other similar exhaustive literature.

5) Similarly, the laboratory Methods should be improved. Please, provide more information and a clearer one.

6) Results. Line 108. Remove > 65.

7) Table 1. Age of patients. The youngest one seems to be 11. Elsewhere you state that the population was 18 or above. Please, remove the under-18 patients and consistently emend the patient total number (less than 822) and all related calculations and percentages.

8) Table 1. p=0.001 reads p = 0.00.

9) Line 121: What is Pearsona?

10) Fig. 1. It should be re-drawn. In order to keep the few under-18 in, the columns do not start at 18 and are irregular. Columns starting with 26 and 60 contain only 4 years. All the other 5 years.

11) Fig. 2. Please, provide more information in the legend.

12) Lines 140-142 and subsequent Fig. 3 and 4. Major point. Either you make yourself properly understood by the average reader about the message that the Foster plot and the ROC curve convey, with a substantial description of this message, or (better) remove Fig. 3 and 4.

13) Fig. 3 and 4. Headings in a language other than English. I suppose Polish. Be clear or remove.

14) Line 162-163. Pearsona! Manna-Whitneya!

15) Table 2. Remove LAR 6 and LAR 24. Not established. Potentially conflicting. Remove in Methods, too.

16) Table 3. Remove.

17) Discussion. Emphasize more what this piece of work ultimately shows: there seems to be little difference between undr-60 and over-60. Particularly, from the immunological point of view.

Make an attempt at providing an explanation for the differences. Speculate on the Ves v1/Ves v5 difference.

18) Good the difference in baseline tryptase level. But you can speculate on it even more.

19) Finally, Fig. 5 shows (with a lot of work done by the Authors) that an impressing proportion of patients, both young and old, have baseline tryptase levels well above the accepted normality range upper limit. Latent mastocytosis? Mast cell activation syndrome? This important piece of evidence is only briefly, very briefly, commented.

Comments on the Quality of English Language

English is acceptable but needs improvement

Author Response

Thank you for the thorough analysis of our article and the valuable comments that have been raised. We will endeavor to incorporate all suggestions and corrections to enhance the quality of the publication. Should there be any specific questions or clarifications needed, please feel free to reach out, as we are more than happy to address any additional inquiries. Once again, we appreciate your time and valuable feedback.

1. Title. The English is overall acceptable (although, in a separate section of the reviewing form, I recommended a moderate editing). Therefore, I suppose that the grammar mistakes in it are due to simple lack of accuracy in the rush of the submission process. “Is the clinical pictures and immunological parameters ...age dependent”. Better: “Are the clinical picture and the immunological parameters...age dependent?

Ad 1. Of course, the correction has been made. Thank you for the title suggestion. "Are clinical presentation and immunological parameters age-dependent?"

2.  Abstract, Introduction and Methods – Database and cohort are acceptable. Here some example of minor inaccuracy (minor but still inaccuracy).

Lines 9-10: comparing patients...to data of patients. You cannot compare a patient to a datum, but you compare a patient to another patient or a set of data to another set of data.

Always leave a space after commas.

Line 20. ...might pose a risk factor... Something can pose a risk. Something else can represent a risk factor.

Lines 14 and 15. IVo...IIIo  The Latin numerals IV and III are usually used instead of 4° and 3°. No need for the small “o” beside them.

The above are only examples of the minor mistakes, mainly stylistic; scattered throughout the manuscript. From now on, I will not remark them one by one. But I warmly invite the Authors to go through the text with attention and emend them all.

Ad.2 The wording of the sentence in line 9-10 has been corrected. Spaces have been added after the commas. In line 20, "risk factor" has been corrected. In lines 14 and 15, "IIIo" has been changed to "III."

3. 60/65 question. This is a major point. It is only source of confusion and adds nothing to the story you want to tell. A recommend removing these two sub-population (i.e: under-65 and over-65). Accordingly, emend all sections of the manuscript.

Ad 3. In response to this point of the review, we would like to present our perspective on the justification for dividing the study group into patients over 60 and over 65 years of age. The definition of a senior citizen is not clear-cut, and depending on the source, the threshold is set at either > 60 or > 65 years of age. The National Health Fund in Poland considers people over 60 as seniors. However, in some European countries, the age limit for seniors is 65, which resulted in the additional classification of the 65+ group and their assessment in the presented study. [Rosenthal T., Naughton B., Williamsa M.: Geriatria, CZELEJ, Lublin, 2009].

It was important for us to demonstrate that regardless of the division criteria (> 60 or > 65 years), both groups did not differ in terms of immunological response or clinical reactivity. The low number of participants in the "older" age groups prevented the assessment of the parameters in the "senior seniors" group. This division also has practical justification. The number of individuals defined as seniors in our clinic is significantly increasing, hence the need to assess this population. We plan to conduct another assessment in patients aged 75 and over, based on the experience that a 5-year interval may not be sufficient to detect any differences.

If these arguments do not seem convincing to you, we are willing to reconsider the evaluation of patients aged 60 and over from all sections of the study.

4. Line 76: Diagnostic procedure. The skin test methodology is described poorly. Moreover, it seems to me that there is a certain confusion with the concentration of the allergen. At line 78, it reads 1 mg/ml (wrong). At line 80, it reads 10-4 g/l. If I am not mistaking, it means 100 micrograms/l (wrong). The usual concentration is 100 micrograms/ml. I recommend improving much this sub-section. The Authors may found inspiration in Corallino M. et al. J Clin Nurs. 2007:16;1256-64. Or other similar exhaustive literature.

Ad 4 The intradermal test was performed according to the EAACI guidelines, which also define the range of diagnostic venom concentrations for intradermal skin testing. In this matter, the EAACI guidelines are consistent with the recommendations of the AAAAI. Below is a quote documenting these recommendations:

"Skin tests are performed by skin prick or intradermal testing. General procedural recommendations are outlined elsewhere (106). Stepwise incremental venom skin tests are recommended. If the patient has a conclusive reaction at a set concentration the test can be stopped. For skin prick test venom concentrations of 0.01–100 μg/ml are usually used. Intradermally a 0.02 ml venom concentration ranging from 0.001 to 1 μg/ml is injected into the volar surface of the forearm."

B. M. Biló, F. Rueff, H. Mosbech, F. Bonifazi, J. N. G. Oude-Elberink, the EAACI Interest Group on Insect Venom Hypersensitivity. Diagnosis of Hymenoptera venom allergy. Allergy 2005. 

5. Similarly, the laboratory Methods should be improved. Please, provide more information and a clearer one.

Ad 5. In the section "Methods" of the document, detailed methods used were described.

6. Results. Line 108. Remove > 65.

Ad 6. Additionally, in the results section, line 108 was modified by removing > 65.

7. Table 1. Age of patients. The youngest one seems to be 11. Elsewhere you state that the population was 18 or above. Please, remove the under-18 patients and consistently emend the patient total number (less than 822) and all related calculations and percentages.

Ad 7. Our analysis included all patients who underwent insect venom immunotherapy in our center, the youngest of whom was 11 years old.

8. Table 1. p=0.001 reads p = 0.00.

Ad 8. A correction was made in table 1 by changing "p=0.001" to "p = 0.00

9. Line 121: What is Pearsona?

Ad 9. Line 121  Our fault. It is incorrect spelling of the Pearson test, also known as the Pearson correlation test.

10. Fig. 1. It should be re-drawn. In order to keep the few under-18 in, the columns do not start at 18 and are irregular. Columns starting with 26 and 60 contain only 4 years. All the other 5 years.

Ad 10. Figure 1 should have been drawn.

11. Fig. 2. Please, provide more information in the legend.

Ad 11. The caption for Figure 2 has been completed.

12. Lines 140-142 and subsequent Fig. 3 and 4. Major point. Either you make yourself properly understood by the average reader about the message that the Foster plot and the ROC curve convey, with a substantial description of this message, or (better) remove Fig. 3 and 4.

Ad 12 Following the Reviewer's suggestion, Figure 3 and 4 have been removed.

13. Fig. 3 and 4. Headings in a language other than English. I suppose Polish. Be clear or remove.

Ad 13. Following the Reviewer's suggestion, Figure 3 and 4 have been removed.

14. Line 162-163. Pearsona! Manna-Whitneya!

Ad14. Lines 162-163. Have been corrected.

15. Table 2. Remove LAR 6 and LAR 24. Not established. Potentially conflicting. Remove in Methods, too.

Ad15 Data regarding LAR has been removed from the methodology and results sections. Consequently, tables 2 and 3 have also been removed.

16. Table 3.

Ad 16. Tab. 2 Will be removed if our explanations regarding the division into 60 and 65 years will not be taken into account

17. Discussion. Emphasize more what this piece of work ultimately shows: there seems to be little difference between undr-60 and over-60. Particularly, from the immunological point of view.

Make an attempt at providing an explanation for the differences. Speculate on the Ves v1/Ves v5 difference.

Ad 17. Text segments related to the main result, indicating no differences in clinical and immunological response between the <60 and >60 age groups, have been added to the discussion.

18. Good the difference in baseline tryptase level. But you can speculate on it even more.

Ad18. Text segments related to this have been added to the discussion

19. Finally, Fig. 5 shows (with a lot of work done by the Authors) that an impressing proportion of patients, both young and old, have baseline tryptase levels well above the accepted normality range upper limit. Latent mastocytosis? Mast cell activation syndrome? This important piece of evidence is only briefly, very briefly, commented.

Ad.19 Text have been added to the discussion 

Round 2

Reviewer 2 Report

Comments and Suggestions for Authors

Dear Authors,

Thank you for addressing my comments in your revisions. I believe the article is now ready for publication.

Best regards,

Comments on the Quality of English Language

Minor editing of English language required.

Author Response

Dear Reviewer

Once again, we sincerely thank you for the detailed and thorough analysis of our article. 

Best regards 

Authors

Reviewer 4 Report

Comments and Suggestions for Authors

At least in the manuscript version that I can see, both title and Fig 1 have remained unchanged. They should be modified, according my previous suggestions and responses by the Authors.

Comments on the Quality of English Language

None

Author Response

Once again, we sincerely thank you for the detailed and thorough analysis of our article.

Question: At least in the manuscript version that I can see, both title and Fig 1 have remained unchanged. They should be modified, according my previous suggestions and responses by the Authors.

Answers:

Regarding your first question, the title has been corrected as suggested.

As for the second question, the age differences in the description on the X-axis of Fig. 1 arose from the fact that it was not possible to divide the age range of 11-79 years into equal intervals, resulting in those discrepancies. We have decided to regenerate Fig. 1 with a division every 10 years, which may enhance readability. If this is not acceptable, we can create a figure with a division every 5 years.
